# Dynamic Neural Representational Decoders for High-Resolution Semantic Segmentation*

**Bowen Zhang,     Yifan Liu,     Zhi Tian,     Chunhua Shen**

The University of Adelaide, Australia

## Abstract

Semantic segmentation requires per-pixel prediction for a given image. Typically, the output resolution of a segmentation network is severely reduced due to the downsampling operations in the CNN backbone. Most previous methods employ upsampling decoders to recover the spatial resolution. Various decoders were designed in the literature. Here, we propose a novel decoder, termed dynamic neural representational decoder (NRD), which is simple yet significantly more efficient. As each location on the encoder's output corresponds to a local patch of the semantic labels, in this work, we represent these local patches of labels with compact neural networks. This neural representation enables our decoder to leverage the smoothness prior in the semantic label space, and thus makes our decoder more efficient. Furthermore, these neural representations are dynamically generated and conditioned on the outputs of the encoder networks. The desired semantic labels can be efficiently decoded from the neural representations, resulting in high-resolution semantic segmentation predictions. We empirically show that our proposed decoder outperforms the decoder in DeeplabV3+ with only $\sim 30\%$ computational complexity, and achieves competitive performance with the methods using dilated encoders with only $\sim 15\%$ computational costs. Experiments on the Cityscapes, ADE20K, and PASCAL Context datasets demonstrate the effectiveness and efficiency of our proposed method.

## 1   Introduction

Semantic segmentation is a fundamental task in computer vision, which requires pixel-level classification on an input image. Fully convolutional networks (FCNs) are the *de facto* standard approaches to this task, which often consist of an encoder and a decoder. We focus on improving the decoder in this work and assume the encoder to be any backbone networks such as ResNet [HZRS16]. Due to the down-sampling layers (*e.g.*, stridden convolutions or pooling) used in these networks, the encoder's outputs are often of much lower resolutions than the input image. Thus, a decoder is used to spatially upsample the output. The decoder can simply be a bilinear upsampling, which directly upscales the low-resolution outputs of encoders to desired resolutions, or it can be a sophisticated network with a stack of convolutions and multi-level features. Note that another approach to tackle the issue of low-resolution outputs is the use of dilation convolution as in DeepLab [CPK$^{+}$17], which balances the need for large receptive fields and maintaining a higher-resolution feature map. The computational cost is significantly heavier introduced by dilation convolutions.

Popular decoders for semantic segmentation include the one in DeepLabV3+ [CPSA17], which fuses the low-level feature maps with $1/4$ resolution of the input image, and RefineNet [LMSR17] which gradually combines multi-level feature maps. A potential drawback of these decodes may be that they

---

*BZ, YL, ZT contributed equally, and are listed alphabetically. CS is the corresponding author (e-mail: `chunhua@me.com`).

35th Conference on Neural Information Processing Systems (NeurIPS 2021).

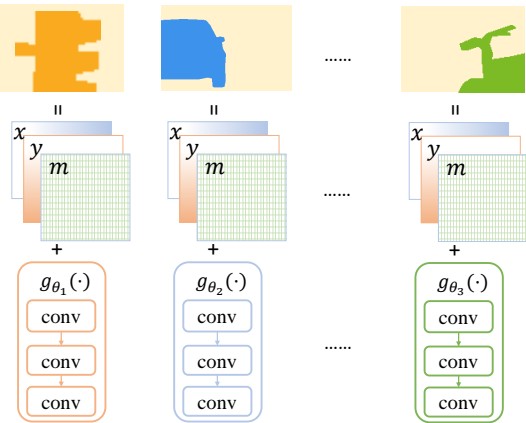

**Figure 1: The overall concept of our neural representations.** The top row is some examples of the semantic label patches. In the neural representations, each patch is represented with a neural network $g_{\theta}(\cdot)$, as shown in the bottom of this figure. The semantic label patch can be recovered by forwarding the coordinate maps (denoted by $x$ and $y$ in the figure) and the guidance maps (*i.e.*, $m$ in the figure) through the network. As stated in our text, using neural representations for these label patches can implicitly take advantage of the smoothness prior in the semantic label patch.

do not explicitly exploit the label dependency, thus being less efficient in recovering the pixel-wise prediction accurately.

Let us consider an $8 \times 8$ local patch on a binary semantic label space, denoted by $P \in \{0, 1\}^{64}$. If we do not consider any structural correlations in the patch, there would be $2^{64}$ possibilities for this local patch. However, it is clear to see, for any natural images, the vast majority of the possibilities never exist in the real label map and only a tiny fraction of them are really possible (see Fig. 1). Considering the redundancy in the labels, most existing decoders that do not explicitly take this into account would be sub-optimal. This motivates us to design a much more effective decoder by exploiting the prior.

A simple approach is dimensionality reduction techniques. As shown in [THSY19], the authors first apply principal component analysis (PCA) to the label patches and compress them into low-dimension compact vectors. Next, the network is required to predict these low-dimension vectors, which are eventually restored into the semantic labels by inverting the PCA process. Their method achieves some success. However, the simplicity and linearity assumption of PCA also limit its performance.

The semantic label masks for natural images are not random and follow some distributions, as shown in Fig. 1. Therefore, a good mask representation/decoder must exploit this prior. For computational efficiency, we also want the decoder to be in a compact form. Thus, we require the prior to be effectively learnable from data. Recently, many works [SMB+20, MON+19, PNM+20] exploit neural networks to represent 3D shapes. The work of [Mit97] found that neural networks enjoy the inductive bias of *smooth interpolation between data points*, which means that for two points of the same label, the neural networks tend to assign the same label to the points between them as well. As a result, we can conclude that the above idea of representing 3D shapes with neural networks can implicitly leverage the local smoothness prior. Therefore, inspired by these works, we can also represent the local patches of semantic labels with neural networks.

To be specific, as shown in Fig. 1, we represent each local label patch by a compact neural network $g_{\theta_i}$ with a few convolution layers interleaved with non-linearities. The semantic labels of a local patch can be obtained by forwarding the corresponding network with $(x, y)$-coordinate maps and a guidance map $m$ (explained later) as inputs. Furthermore, the parameters $\theta$ of these neural networks, which represent the local label patches, can be dynamically generated with the encoder network in FCNs, and each location on the encoder's output feature maps is responsible for generating the parameters of the neural network representing the specific local label patch surrounding it. The dynamic network makes it possible to incorporate the neural representations into the conventional encoder-decoder architectures and enables a compact design of the decoder, resulting in an end-to-end trainable framework. This avoids the separable learning process as done in [THSY19].

Thus, our method is termed *dynamic neural representation decoder* (NRD) for semantic segmentation. We summarize our main contributions as follows.

- We propose a novel decoder that is effective and compact for semantic segmentation, to recover the spatial resolutions. For the first time, we represent the local label patches using neural networks and make use of dynamic convolutions to parametrize these neural networks.

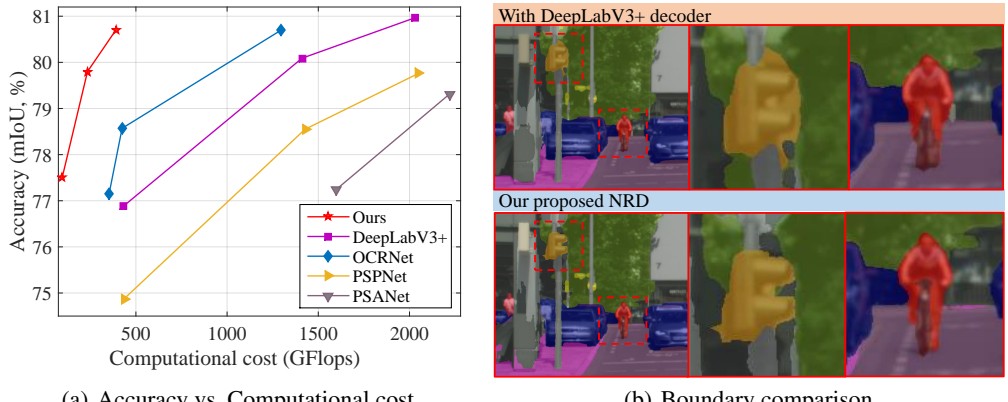

| (a) Accuracy vs. Computational cost | (b) Boundary comparison |

**Figure 2:** (a) Accuracy vs. computational cost on the validation set of Cityscapes. Our proposed NRD can achieve a better trade-off. (b) Comparison between our proposed NRD and the decoder in DeepLabV3+ [CZP$^+$18]. We can see that NRD is capable of generating improved boundaries.

- Different from previous methods, which often neglect the redundancy in the semantic label space, our proposed decoder NRD can better take advantage of the redundancy, and thus it is able to achieve *on par* or improved accuracy with significantly reduced computational cost. As shown in Fig. 2(a), we achieve a better trade-off between computational cost and accuracy compared to previous methods.

- Compared with the decoder used in classic encoder-decoder model DeeplabV3+ [CZP$^+$18], we achieve an improvement of $0.9\%$ mIoU on the Cityscapes dataset with less than $30\%$ computational cost. Moreover, on the trimaps, where only the pixels near the object boundaries are evaluated, a $1.8\%$ improvement can be obtained. This suggests that NRD can substantially improve the quality of the object boundaries.

  Moreover, NRD is even more significant than some methods that use dilated encoders, which usually require $4\times$ more computational cost than ours with similar accuracy. For example, NRD achieves $46.09\%$ mIoU on the competitive ADE20K dataset, which is comparable to that of DeepLabV3+ with a dilated encoder ($46.35\%$) but with only $30\%$ computational cost. We also benchmark our method on the Pascal Context dataset and show excellent performance with much less computational cost.

## 1.1 Related Work

**Neural network representations.** Recently, many works [SMB$^+$20, MON$^+$19, PNM$^+$20] exploit neural networks to represent 3D shapes, which follow the idea that a 3D shape can be represented with a classification model and the 3D shape can be restored by forwarding the 3D coordinates through the classification network. These methods can be viewed as representing the point cloud data with the neural network's parameters.

**Dynamic filter networks.** Different from traditional convolutions whose filters are fixed during inference once learned, the filters are dynamically generated by another network (namely, the controller). This idea was proposed by [JDBTvG16], which enlarges the capacity of the network and captures more content-dependent information such as contextual information. Recently, CondInst [TSC20] makes use of dynamic convolutions to implement the dynamic mask heads, which are used to predict the masks of individual instances. In this work, we follow in this vein for a different purpose, which is to dynamically generate the parameters of the networks representing local label masks so as to produce high-resolution semantic segmentation results.

**Encoder-decoder network architectures.** The encoder-decoder architecture is widely used to solve the semantic segmentation task, and almost all the mainstream semantic segmentation methods can be categorized into this family. Typically, the encoder gradually reduces the resolution of feature maps and extracts semantic features, while the decoder is applied to the output features of the encoder

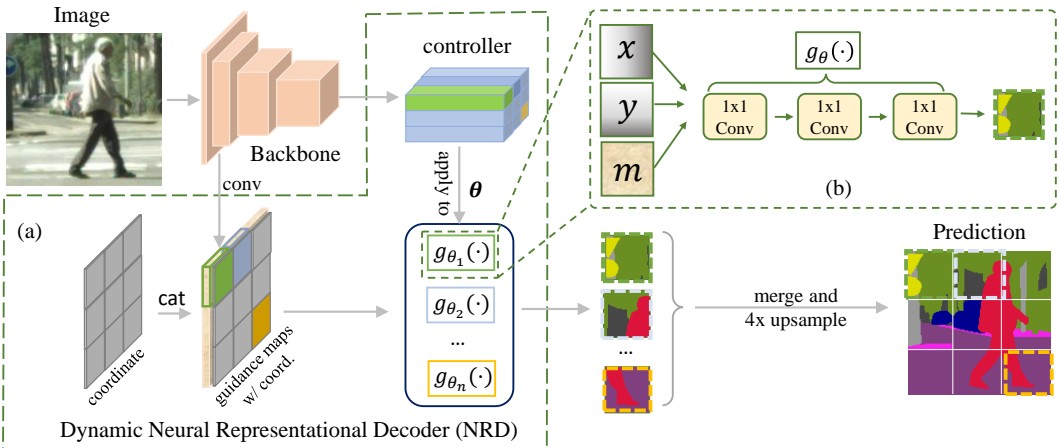

**Figure 3: The framework of our proposed decoder.** (a) The proposed NRD Module. (b) The details of one of the representational networks $g_\theta(\cdot)$. As we can see, we apply the controller to the encoder's output feature maps and generates the parameters $\theta$ of the representational networks. Note that each location on the encoder's output feature maps generate a different set of parameters, which correspond to the representational network of the local patch surrounding the location. Thus we have $H' \times W'$ sets of parameters in total, where $H'$ and $W'$ are the height and width of the encoder's output, respectively. Afterwards, the representational networks are fed the $(x, y)$-coordinate maps and guidance maps $m$ to predict semantic label patches. The guidance maps are generated by applying convolutions to low-level feature maps. We use the same low-level feature maps here as in DeepLabv3+. Finally, these patches are merged into the desired high-resolution segmentation results.

to decode the desired semantic labels and recover the spatial resolution. Our work here focuses on the decoder.

The most commonly used bilinear upsampling can be viewed as the simplest decoder, which assumes that the semantic label maps are smooth to a large extent and the linear interpolation is sufficient to approximate them. Thus, using bilinear upsampling here is effective when the semantic label maps are simple, but the performance is not satisfactory if the label maps are complicated. DeconvNet [NHH15] introduces deconvolutional layers in its decoder to step-by-step recover the resolution of the prediction, which can result in much better performance. UPerNet [XLZ+18] uses an FPN-like structure to fuse feature maps of different scales, and obtains high-resolution feature maps. DeepLabv3+ [CZP+18] designs an effective decoder module that makes use of both encoder-decoder structure and dilation/atrous convolution, which is still one of the most competitive segmentation methods to date, especially in the trade-off between accuracy and computation complexity. CARAFE [WCX+19] first upsamples feature maps with parameter-free methods and then applies a learnable content-aware kernel mask to the upsampled feature maps. Thus far, despite achieving some success, we believe that there is much room for improvement in terms of taking full advantage of label space prior and designing highly effective and compact decoders for semantic segmentation. The proposed NRD attempts to narrow this gap.

## 2 Our Method

### 2.1 Overall Architecture

Given an input image $I \in \mathbb{R}^{H \times W \times 3}$, the goal of semantic segmentation is to provide the pixel-level classification score map of the shape of $H \times W \times C$, where $C$ equals the number of categories to be classified into. As mentioned above, mainstream semantic segmentation methods are often based on encoder-decoder architectures. We also follow this line. Fig. 3 shows the overall framework of the proposed model for semantic segmentation.

Our work focuses on the decoder part, and thus we simply make our encoder the same as DeeplabV3+ [CZP+18]. The encoder consists of a CNN backbone (*e.g.*, ResNet) and some optional modules such as ASPP [CPSA17], which can enhance the output features. By forwarding an input image $I \in \mathbb{R}^{H \times W \times C}$ through the encoder, it generates feature maps with the shape of $H/r \times W/r \times D$,

where $D$ is the number of the channels of the feature maps and $r$ is the downsampling ratio of the encoder.

The downsampling ratio is determined by the down-sampling operators in the encoder and can be adjusted by reducing the stride of these down-sampling operators. Dilated convolutions are often used to compensate for the reduction of receptive fields after reducing the strides, with the price of computation overhead. An encoder that reduces the strides and uses dilation convolutions is often referred to as a *dilated encoder*. For example, an encoder based on the standard ResNet backbone produces the feature maps with $r = 32$. Most methods [YCW19, CZP$^+$18, ZDS$^+$18] dilate the encoder and reduce $r$ to $16$ or $8$. By using the dilated encoder, these methods can output higher-resolution results while the dilated encoder would significantly increase the computational cost. In our work, we do not dilate the encoders (*e.g.*, using $r = 32$) for faster computation and our proposed NRD is expected to better predict the semantic mask at a high resolution.

Let us denote the encoder's output feature maps by $F \in \mathbb{R}^{\frac{H}{32} \times \frac{W}{32} \times D}$, whose resolution is $1/32$ of the input image and the desired semantic label map (*i.e.*, the final results). Thus, we make each spatial location on $F$ responsible for a $32 \times 32$ local patch surrounding the location and predict the local label map of the patch with our proposed NRD. Finally, the label maps of these patches are merged into the full-resolution segmentation results.

## 2.2 Dynamic Neural Representational Decoders (NRD)

In this section, we provide the details of our NRD and how we generate the parameters for it. The core idea here is to make use of a neural network to represent a local label patch. Thus, given a ground-truth semantic label map $Y \in \{0, 1, ..., C-1\}^{H \times W}$, following the convention, we first convert it to the one-hot label map $Y' \in \{0, 1\}^{H \times W \times C}$, where $C$ is the number of classes. Next, $Y'$ is divided into a number of $H' \times W'$ local patches, and let $P \in \mathbb{R}^{r \times r \times C}$ be one of the patches, where $H'$ and $W'$ are the height and width of the encoder's outputs and $r$ is 32 in our work. Let us take Cityscapes as an example, and thus we have $C = 19$ and $P \in \mathbb{R}^{32 \times 32 \times 19}$. Next, a compact network $g_{\boldsymbol{\theta}}(\cdot)$ is designed to represent the local mask patch $P$, as shown in Fig. 1. To be specific, in our experiment, $g_{\boldsymbol{\theta}}(\cdot)$ is composed of three $1 \times 1$ convolutions interleaved with the non-linearity ReLU. Except for the input and output channels, all the hidden layers in $g_{\boldsymbol{\theta}}(\cdot)$ have 16 channels. The output channels of $g_{\boldsymbol{\theta}}(\cdot)$ is equal to the number of classes (*i.e.*, $C$).

To recover the local patch $P$, we apply $g_{\boldsymbol{\theta}}(\cdot)$ to a $(x, y)$-coordinate map $Q = [0 : \frac{1}{s} : 1] \times [0 : \frac{1}{s} : 1] \in \mathbb{R}^{s \times s \times 2}$, where $[0 : \frac{1}{s} : 1]$ is the range from 0 to 1 with step $\frac{1}{s}$ ($s$ is the desired upsampling rate, being 8 in this work ) and '$\times$' means the Cartesian multiplication. Since $g_{\boldsymbol{\theta}}(\cdot)$ is composed of $1 \times 1$ convolutions, the outputs of $g_{\boldsymbol{\theta}}(\cdot)$ also have size $s \times s$ and can be denoted as $G \in \mathbb{R}^{s \times s \times 19}$. As shown in Fig. 3-(b), $g_{\boldsymbol{\theta}}(\cdot)$ takes guidance maps $m \in \mathbb{R}^{s \times s \times C_m}$ as additional inputs. $m$ is generated by applying two convolutional layers to the low-level feature maps, which reduce the channels of the feature maps to $C_m$, being 16 in this work. We use the same low-level feature maps as in DeepLabv3+, whose resolutions are $1/4$ of the input image. Afterward, a bilinear upsampling is used to upscale $G$ by 4 times to obtain $P' \in \mathbb{R}^{32 \times 32 \times 19}$. Next, we compute the loss between $P'$ and $P$, which, through the back-propagation, adjusts the network's parameter $\boldsymbol{\theta}$ so that $P'$ is as similar to $P$ as possible. In this way, the network parameters $\boldsymbol{\theta}$ can be viewed as the representation of the local semantic label patch $P$. Although it is possible to remove the bilinear upsampling here and, by increasing the resolution of $Q$ and $m$, to make the network directly output the desired resolution $32 \times 32$, we do not adopt this because using bilinear is sufficient when the upsampling factor is small (*e.g.*, being 4 here). We note that in the above case the network $g_{\boldsymbol{\theta}}(\cdot)$ has 899 parameters in total ($\#weights = (2 + 16) \cdot 16(conv1) + 16 \cdot 16(conv2) + 16 \cdot 19(conv3)$ and $\#biases = 16(conv1) + 16(conv2) + 19(conv3)$).

As shown in previous works [THSY19, WCY$^+$18], each location on the encoder's output feature maps can encode the information of the local patch surrounding it. Therefore, inspired by dynamic filter networks [YWP$^+$18b], we can use the decoder's output features at each location to dynamically generate the parameters of the representational network for the label patch of the location. To be specific, given the encoder's output feature maps $F \in \mathbb{R}^{H' \times W' \times D}$, where $H' = H/32$, $W' = W/32$ and $D$ are height, width, and the number of channels of $F$.

**Controller.** We apply a $3 \times 3$ convolution with $512$ channels, which is followed by a $1 \times 1$ convolution to generate the parameters $\boldsymbol{\theta}$ (shown as the 'controller' in Fig. 3). The number of output channels of

the convolution is equal to the number of parameters in $\boldsymbol{\theta}$. The generated parameters are then split and reshaped into the weights and biases in $g_{\boldsymbol{\theta}}(\cdot)$, and then $g_{\boldsymbol{\theta}}(\cdot)$ is forwarded to obtain the semantic prediction $P'$. $P'$ is supervised by the ground-truth label patch $P$, making the whole framework end-to-end trainable. The overall architecture is shown in Fig. 3.

## 3   Experiments

The proposed model is evaluated on three semantic segmentation benchmarks. The performance is measured in terms of intersection-over-union averaged across the present classes (mIoU). We also evaluate the performance near the object boundaries by calculating mIoU on the trimap following [CZP$^+$18]. We evaluate our method on the following benchmarks.

**ADE20K** [ZZP$^+$17] is a dataset that contains more than 20K images exhaustively annotated with pixel-level annotation. It has $20,210$ images for training and $2,000$ images for validation. The number of categories is 150.

**PASCAL Context** [MCL$^+$14] is a dataset with $4,998$ images for training and $5,105$ images for validation. We use default settings in [MMS20] that chose the most frequent 59 classes plus one background class (60 classes in total) as the targets.

**Cityscapes** [COR$^+$16] is a benchmark for semantic urban scene parsing. The training, validation and test splits contain $2,975$, $500$ and $1,525$ images with fine annotations, respectively. All images from this dataset are $1024 \times 2048$ pixels in size.

**Implementation details.** We use ResNet-50 and ResNet-101 [HZRS16] as our backbone networks and initialize them with the ImageNet pre-trained weights. The training and testing settings as well as data augmentations inherit the default settings in [MMS20] unless specified. Specifically, for all datasets, we use 'poly' as our learning policy. The initial learning rate is set at $0.01$, the weight decay is set to $0.0005$ for Cityscapes and ADE20K. For PASCAL Context, the initial learning rate is $0.004$ and the weight decay is $0.0001$. We train ADE20K, PASCAL-Context and Cityscapes for 160k, 80k and 80k iterations, with the crop size of $512 \times 512$, $480 \times 480$ and $512 \times 1024$, respectively. The training and testing environment is on a workstation with four Volta 100 GPU cards. For test time augmentation, we employ the horizontal flip and multi-scale inference. The scale factors are $\{0.5, 0.75, 1.0, 1.25, 1.5, 1.75\}$.

### 3.1   Ablation Study

In this section, we conduct the ablation study to show the effectiveness of our proposed NRD. Here, we first compare NRD with the decoder of DeeplabV3+ [CZP$^+$18] since it is widely-used in practice. Then, we compare with other decoder methods. Note that when these methods are compared, we use the same encoder for them. Finally, we investigate the hyper-parameters of our model design.

**Compared to the DeepLabV3+ decoder.** Since we do not use dilation convolutions in our encoder, we also remove the dilation in the DeeplabV3+ encoder for a fair comparison. The results are shown in Table 1. As shown in the table, with exactly the same settings, NRD outperforms the decoder in DeeplabV3+ by $0.9\%$ mIoU on the Cityscpaes `val.` split with less than $^1/_3$ computational cost ($20.4$ vs. $76.4$ GFlops), and the total computational cost including the encoder and decoder is reduced from $290.6$ to $234.6$ GFlops. In addition, on the trimap, NRD is $1.8\%$ mIoU better than the DeeplabV3+ decoder, which suggests that our method is able to produce boundaries of higher quality.

**Compared to the bilinear decoder.** We also compare our method with the simplest decoder which uses a $1 \times 1$ convolution to map the outputs of the encoder to the desired segmentation predictions and then simply uses the bilinear upsampling to upscale the predictions to the desired resolutions. Again, both encoders' output resolutions are $^1/_{32}$ of the input image. To make a fair comparison, we also remove the guidance map in NRD (*e.g.*, the low-level features). Thus, only coordinate maps are taken as the input of NRD. As shown in Table 1, NRD surpasses the bilinear decoder by a large margin ($+3.5\%$ mIoU). Note that although NRD has a higher computational cost than the bilinear decoder ($1.3$ vs. $2.6$ GFlops), the overall computational cost is almost the same ($215.5$ vs. $216.8$ GFlops) as most of the computational cost is in the encoder. Additionally, the mIoU on the trimap is improved by $5.4\%$.

**Table 1: Our proposed NRD vs. the DeepLabV3+ decoder and bilinear decoder** on the Cityscapes `val`. split. All models use the same encoder and are trained with 84K iterations and $512 \times 1024$ crop size. The GFlops is measured with the original image size $1024 \times 2048$. All the GFlops in this paper are measured at single scale inference. GFlops$^{\text{dec}}$ indicates the GFlops for decoders only.

| Method | Backbone | Low-level | GFlops$^{\text{dec}}$ | GFlops | mIoU (%) | Trimap mIoU (%) |
|---|---|---|---|---|---|---|
| Decoder | ResNet-50 | stage2 | 76.4 | 290.6 | 78.9 | 49.8 |
| NRD (**Ours**) | ResNet-50 | stage2 | 20.4 | 234.6 | **79.8** (+0.9) | **51.6** (+1.8) |
| Bilinear decoder | ResNet-50 | None | 1.3 | 215.5 | 74.7 | 41.2 |
| NRD (**Ours**) | ResNet-50 | None | 2.6 | 216.8 | **78.2** (+3.5) | **46.6** (+5.4) |

**Table 2:** Comparison of different up-sampling methods using ResNet50 as backbones on the Cityscapes `val`. split. All methods are trained for $84k$ iterations. The GFlops is measured at single scale inference with a crop size of $1024 \times 2048$. The proposed NRD outperforms previous decoders.

| Method | GFlops | Params | mIoU (%) |
|---|---|---|---|
| CARAFE | 203.0 | 36.3 | 72.1 |
| DUC | 336.1 | 110.8 | 74.7 |
| **NRD** (**Ours**) | 203.2 | 36.6 | **75.0** |

**Table 3:** Ablation results on the Cityscapes validation set. $C_r$ is the number of channels of the $1 \times 1$ convolutions in $g_{\boldsymbol{\theta}}(\cdot)$. $C_m$ is the number of channels of the guidance map. The accuracy is not very sensitive to these parameters and in general 16 channels for both $C_r$, $C_m$ lead to marginally better results.

| | NRD variants | | | | | |
|---|---|---|---|---|---|---|
| $C_r$ | 8 | **16** | 32 | 16 | **16** | 16 |
| $C_m$ | 16 | **16** | 16 | 8 | **16** | 32 |
| mIoU | 79.4 | **79.8** | 79.6 | 79.5 | **79.8** | 79.0 |

**Compared to other decoder methods.** We also compare NRD with some other decoder methods. ResNet-50 is used as the backbone and we do not use the dilated encoders in all these methods. The results are shown in Table 2. As shown in the table, compared to CARAFE [WCX$^+$19], we improve the mIoU on Cityscapes from 72.1% to 75.0% with similar computational complexity (203.0 vs. 203.2 GFlops) and the number of parameters. In addition, compared to DUC [WCY$^+$18], which outputs multiple channels and use the "depth-to-space" operation to increase the spatial resolutions, our NRD is superior to it (75.0% vs. 74.7% mIoU) with only 60% computational complexity (203.2 vs. 336.1 GFlops) and $\sim$33% parameters.

**Ablation study of architectures of NRD.** Here, we investigate the hyper-parameters of our NRD. Table 3 shows the performance as we vary the number of channels $C_r$ of the representational network $g_{\boldsymbol{\theta}}(\cdot)$. As we can see in the table, the performance is not very sensitive to the number of channels (within 0.4% mIoU). We also experiment by varying the number of channels $C_m$ of the guidance map $m$. As shown in Table 3, using $C_m = 16$ can result in slightly better performance than $C_m = 8$ (79.8% vs. 79.5% mIoU), but increasing $C_m$ to 32 cannot improve the performance further.

Table 4 shows the effect of the inputs to the representational network $g_{\boldsymbol{\theta}}(\cdot)$. As we can see, if no guidance maps are given and $g_{\boldsymbol{\theta}}(\cdot)$ only takes as input the coordinate maps, NRD can already achieve descent performance (78.2% mIoU), which is already much better than the bilinear decoder as shown in Table 1. In addition, if $g_{\boldsymbol{\theta}}(\cdot)$ only takes the guidance map as input, NRD can achieve similar performance (78.3% mIoU). However, it can be seen that there is a significant improvement on the trimap mIoU (+3.9% mIoU), which suggests that the guidance map plays an import role in preserving the details. Finally, if both the coordinate maps and the guidance maps are used, NRD can achieve the best performance (79.8% mIoU).

## 3.2 Comparisons with state-of-the-art methods

In this section, we compare our method with other state-of-the-art methods on three dataset: ADE20K, PASCAL-Context and Cityscapes.

**ADE20K.** Table 5 shows the comparisons with state-of-the-art methods on ADE20K. Our method achieves 45.62% in terms of mIoU with ResNet-101 as the backbone. It is 0.95% better than the recent SFNet [LYZ$^+$20], with the same ResNet-101 backbone. Besides, due to the strong ability of NRD to recover the spatial information, we do not need to use the multi paths complex decoder as

**Table 4:** NRD results with various inputs to the representational network $g_\theta(\cdot)$. 'Guidance map': use of the guidance map as the inputs to the representational network or not; 'Coord. map': use of the coordinate maps or not. The experimental results are evaluated on the Cityscapes `val.` split. We can see that the guidance map is critial to the segmentation accuracy at the object boundaries (see the trimap mIoU).

| Method | Guidance map | Coord. map | mIoU (%) | Trimap mIoU (%) |
|---|---|---|---|---|
| NRD | | ✓ | 78.2 | 46.6 |
| NRD | ✓ | | 78.3 | 50.5 |
| NRD | ✓ | ✓ | **79.8** | **51.5** |

**Table 5:** Experiment results on the ADE20K `val.` split. The GFlops is measured at single scale inference with a crop size of $512\times512$. 'ms' means that mIoU is calculated using multi-scale inference. $^*$ means that results are re-implemented by [MMS20]. Note that compared to the DeepLabv3+, we achieve similar performance (46.09% vs. 46.35% mIoU) with $\sim 30\%$ computational complexity (87.9 vs. 255.1 GFlops). Speed (frames per second, FPS) is measured with the same input size as the single scale inference on a RTX 3090 GPU.

| Method | Backbone | Dilated encoder | GFlops | mIoU (%) | mIoU 'ms' (%) | FPS |
|---|---|---|---|---|---|---|
| PSPNet [ZSQ$^+$17] | ResNet-50 | ✓ | 178.8 | 41.68 | 42.78 | 30.01 |
| PSANet [ZZL$^+$18] | ResNet-50 | ✓ | 194.8 | 41.92 | 42.97 | 25.60 |
| EncNet [ZDS$^+$18] | ResNet-50 | ✓ | >100 | - | 41.11 | 33.34 |
| CFNet [ZZWX19] | ResNet-50 | ✓ | >100 | - | 42.87 | - |
| RGNet [YLG$^+$20] | ResNet-50 | ✓ | >100 | - | 44.02 | - |
| CPNet [YWG$^+$20] | ResNet-50 | ✓ | 208.6 | 43.92 | 44.46 | 27.96 |
| DeepLabv3+$^*$ [CZP$^+$18] | ResNet-50 | ✓ | 177.5 | 43.95 | 44.93 | 29.23 |
| PSPNet [ZSQ$^+$17] | ResNet-101 | ✓ | 256.4 | 41.96 | 43.29 | 20.25 |
| PSPNet [ZSQ$^+$17] | ResNet-269 | ✓ | - | 43.81 | 44.94 | - |
| PSANet [ZZL$^+$18] | ResNet-101 | ✓ | 272.5 | 42.75 | 43.77 | 18.11 |
| EncNet [ZDS$^+$18] | ResNet-101 | ✓ | >180 | - | 44.65 | 21.89 |
| CFNet [ZZWX19] | ResNet-101 | ✓ | >180 | - | 44.89 | - |
| CCNet [HWH$^+$19] | ResNet-101 | ✓ | >180 | - | 45.22 | - |
| ANLNet [ZXB$^+$19] | ResNet-101 | ✓ | >180 | - | 45.24 | - |
| GFFNet [LZH$^+$20] | ResNet-101 | ✓ | >180 | - | 45.33 | - |
| DMNet [HDQ19] | ResNet-101 | ✓ | >180 | - | 45.5 | - |
| RGNet [YLG$^+$20] | ResNet-101 | ✓ | >180 | - | 45.8 | - |
| CPNet [YWG$^+$20] | ResNet-101 | ✓ | 286.3 | 45.39 | 46.27 | 18.25 |
| DeepLabv3+$^*$ [CZP$^+$18] | ResNet-101 | ✓ | 255.1 | **45.47** | **46.35** | 19.74 |
| SFNet [LYZ$^+$20] | ResNet-50 | | 83.2 | - | 42.81 | 27.64 |
| SFNet [LYZ$^+$20] | ResNet-101 | | 102.7 | - | 44.67 | 21.95 |
| EfficientFCN [LHZ$^+$20] | ResNet-101 | | 60.5 | - | 45.28 | 53.87 |
| OCRNet [YCW19] | HRNetV2-W48 | | 164.8 | - | 45.66 | 16.39 |
| NRD (**Ours**) | ResNet-101 | | **49.0** | 44.01 | 45.62 | **54.06** |
| NRD (**Ours**) | ResNeXt-101 | | 87.9 | **44.34** | **46.09** | 34.88 |

in SFNet and thus our method only spends $50\%$ computational cost of SFNet. Our method is also better than other methods with dilated encoders, including DMNet [HDQ19], ANLNet [ZXB$^+$19], CCNet [HWH$^+$19] and EncNet [ZDS$^+$18], and needs only $20\% \sim 30\%$ computational cost of these methods. Additionally, by using a larger backbone ResNext-101, our performance can be further improved to $46.09\%$ mIoU. Note that even with the larger backbone, our method still has much lower computational complexity than other methods with dilated encoders. As a result, we can achieve competitive performance among state-of-the-art methods with significantly less computational cost.

**PASCAL-Context.** Table 6 shows the results on the PASCAL-Context dataset. We follow HR-Net [SZJ$^+$19] to evaluate our method and report the results under 59 classes (without background) and 60 classes (with background). Our methods achieve $54.1\%$ (59 classes) and $49.0\%$ (60 classes) mIoU. The results are even better than the sophisticated high-resolution network HRNet with $\sim 50\%$ computational complexity ($42.9$ vs. $82.7$ GFlops). Note that HRNet stacks some hourglass networks and is much complicated than ours. Our method also achieves better results with less computational cost than other methods, as shown in the table.

**Table 6:** Semantic segmentation results on the PASCAL-Context `val.` split. $mIoU_{59}$: mIoU averaged over 59 classes (without background). $mIoU_{60}$: mIoU averaged over 60 classes (59 classes plus background). Both metrics were used in the literature; and we report both for thorough comparisons. Following published methods, we report the results with multi-scale inference (denoted by 'ms'). The GFlops is measured at single scale inference with a crop size of $480 \times 480$. 'Dilated-$*$': using dilated encoders.

| Method | Backbone | GFlops | $mIoU_{59}$ (ms) | $mIoU_{60}$ (ms) | FPS |
|---|---|---|---|---|---|
| FCN-8s [LSD15] | VGG-16 | - | - | 35.1 | - |
| HO-CRF [AJZT16] | - | - | - | 41.3 | - |
| Piecewise [LSvR16] | VGG-16 | - | - | 43.3 | - |
| DeepLab-v2 [CPK+17] | Dilated-ResNet-101 | - | - | 45.7 | - |
| RefineNet [LMSR17] | ResNet-152 | - | - | 47.3 | - |
| UNet++ [ZSTL18] | ResNet-101 | - | 47.7 | - | - |
| PSPNet [ZSQ+17] | Dilated-ResNet-101 | 157.0 | 47.8 | - | 22.45 |
| Ding *et al.* [DJS+18] | ResNet-101 | - | 51.6 | - | - |
| EncNet [ZDS+18] | Dilated-ResNet-101 | 192.1 | 52.6 | - | 24.66 |
| HRNet [SZJ+19] | HRNetV2-W48 | 82.7 | 54.0 | 48.3 | 19.90 |
| GFFNet [LZH+20] | Dilated-ResNet-101 | - | 54.3 | - | - |
| EfficientFCN [LHZ+20] | ResNet-101 | 52.8 | 55.3 | - | 47.8 |
| OCRNet [YCW19] | HRNetV2-W48 | 143.9 | 56.2 | - | 17.11 |
| NRD (**Ours**) | ResNet-101 | **42.9** | **54.1** | **49.0** | **49.60** |

**Table 7:** Experiment results on the Cityscapes `test` split. 'ms' means that mIoU is calculated using multi-scale inference. The GFlops is measured at single scale inference with a crop size of $1024 \times 2048$.

| Method | Backbone | GFlops | mIoU | mIoU (ms) | FPS |
|---|---|---|---|---|---|
| PSPNet [ZSQ+17] | Dilated-ResNet-101 | 2049.0 | - | 78.4 | 3.36 |
| AAF [KHLY18] | Dilated-ResNet-101 | >1500 | - | 79.1 | - |
| DFN [YWP+18b] | Dilated-ResNet-101 | >1500 | - | 79.3 | - |
| PSANet [ZZL+18] | Dilated-ResNet-101 | 2218.6 | - | 80.1 | 2.86 |
| RGNet [YLG+20] | Dilated-ResNet-101 | >1500 | - | 81.5 | - |
| DeepLabV3+ [CZP+18] | Dilated-ResNet-101 | 2032.3 | - | 81.3 | - |
| DANet [FLT+19] | Dilated-ResNet-101 | 2214.7 | - | 81.5 | - |
| GFFNet [LZH+20] | Dilated-ResNet-101 | >1500 | - | 82.3 | - |
| BiSeNet [YWP+18a] | ResNet-101 | >360 | - | 78.9 | - |
| SFNet [LYZ+20] | ResNet-18 | 243.9 | 78.9 | 79.5 | 13.6 |
| HRNet [SZJ+19] | HRNetV2-W48 | 748.7 | - | 81.6 | 7.86 |
| SFNet [LYZ+20] | ResNet-101 | 821.2 | - | **81.8** | 4.62 |
| NRD (**Ours**) | ResNet-50 | 234.6 | 78.9 | 80.0 | 18.17 |
| NRD (**Ours**) | ResNet-101 | 390.0 | 79.3 | 80.5 | 12.23 |

**Cityscapes.** Table 7 shows the performance of our method on the Cityscapes `test` split. We train our model with the `trainval` split and only the fine annotations. As we can see, the proposed method can achieve competitive performance with much less computational complexity. Compared to the recent RGNet [YLG+20], our method achieves comparable performance with less than $30\%$ computational cost (>1500 vs. 390.0 GFlops). Our method also has competitive performance with SFNet less than $50\%$ computational complexity (821.2 vs. 390 GFlops). In addition, it is worth noting that our method based on ResNet-50 can have better performance than ResNet-18 based SFNet ($79.5\%$ vs. $80.0\%$ mIoU) while having even less computational complexity (234.6 vs. 243.9 GFlops). This suggests that our proposed method has a better speed-accuracy trade-off as shown in Fig. 2(a).

## 4 Conclusion

We have proposed a compact yet very effective decoder, termed Neural Representational Decoders (NRD), for the semantic segmentation task. For the first time, we use the idea of neural representations for designing the segmentation decoder, which is able to better exploit the structure in the semantic

segmentation label space. To implement this idea, we dynamically generate the neural representations with dynamic convolution filter networks so that the neural representations can be incorporated into the standard encoder-decoder segmentation architectures, enabling end-to-end training. We show on a number of semantic segmentation benchmarks that our method is highly efficient and achieves state-of-the-art accuracy. We believe that our method can be a strong decoder in high-resolution semantic segmentation and may inspire other dense prediction tasks such as depth estimation and super-resolution. Last but not the least, our method still has some limitations. One of the limitations is that the dynamic filter networks have not been well-supported in some mobile devices, which might restrict the applicability of this method.

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
