**Table 1:** Accuracy and computational costs of different networks on Cityscapes `val.` split. The model mIoU is measured at single scale inference. The GFlops is measured at single scale inference with a crop size of $1024 \times 2048$. Note that comparison results are quoted from [1] which provides re-implementation of all the models in the table.

| Method | backbone | GFlops | mIOU (%) |
|---|---|---|---|
| PSANet | Dilated-ResNet-50 | 1597.15 | 77.24 |
| PSANet | Dilated-ResNet-101 | 2218.62 | 79.31 |
| PSPNet | Dilated-ResNet-18 | 434.11 | 74.87 |
| PSPNet | Dilated-ResNet-50 | 1427.47 | 78.55 |
| PSPNet | Dilated-ResNet-101 | 2048.95 | 79.76 |
| DeepLabv3+ | Dilated-ResNet-18 | 433.9 | 76.89 |
| DeepLabv3+ | Dilated-ResNet-50 | 1410.86 | 80.09 |
| DeepLabv3+ | Dilated-ResNet-101 | 2030.3 | **80.97** |
| OCRNet | HRNetV2p-W18-S | 353.47 | 77.16 |
| OCRNet | HRNetV2p-W18 | 424.29 | 78.57 |
| OCRNet | HRNetV2p-W48 | 1296.77 | 80.7 |
| NRD (**Ours**) | ResNet-18 | **95.7** | 77.5 |
| NRD (**Ours**) | ResNet-50 | 234.6 | 79.8 |
| NRD (**Ours**) | ResNet-101 | 390.0 | 80.7 |

**Table 2:** Experiment results on the ADE20K *val.* split. The GFlops is measured at single scale inference using crop sizes provided in the table. 'ms' means that mIoU is calculated using multi-scale inference.

| Method | Backbone | Crop size | GFlops | mIoU (%) | mIoU 'ms' (%) |
|---|---|---|---|---|---|
| SegFormer | MiT-B1 | $512 \times 512$ | 15.9 | 42.2 | 43.1 |
| NRD | MiT-B1 | $512 \times 512$ | 14.7 | **42.9** | **44.3** |
| SegFormer | MiT-B2 | $512 \times 512$ | 62.4 | 46.5 | 47.5 |
| NRD | MiT-B2 | $512 \times 512$ | 24.5 | **46.8** | **48.2** |
| SegFormer | MiT-B5 | $640 \times 640$ | 183.3 | 51.0 | 51.8 |
| NRD | MiT-B5 | $640 \times 640$ | 124.2 | **51.2** | **51.9** |

# A   Appendix

In this section, we show more evaluation results to demonstrate the effectiveness of the proposed NRD.

## A.1   Additional Evaluation Results

**Evaluation on Cityscapes.** Table 1 shows the comparison of mIOU performance for different methods on Cityscapes `val` split. The GFlops and the mIOU performance are all measured in the same structure implemented in [1]. NRD performs better in terms of accuracy-computation trade-off. NRD usually has the best performance among methods that are of similar computational cost. Even if compared with the methods that use dilated backbones which have 5 times of the computational cost, NRD is still competitive in accuracy.

**Evaluation on ADE20K.** We further use the recent transformer based backbone SegFormer [2] as the encoder to show the ability of the proposed method. The experiments are conducted on the ADE20K dataset. We replace the lightweight all-MLP decoder proposed by Segformer with NRD and follow all training settings in [2]. The results show that using the exactly same backbone, the performance of NRD is competitive with much lighter computation.

## A.2   More Visualization Results

**Comparison with bilinear upsampling.** Fig. 1 shows the result comparison between the bilinear decoder and NRD decoder. Note that here, the results are generated from feature maps that are $1/32$

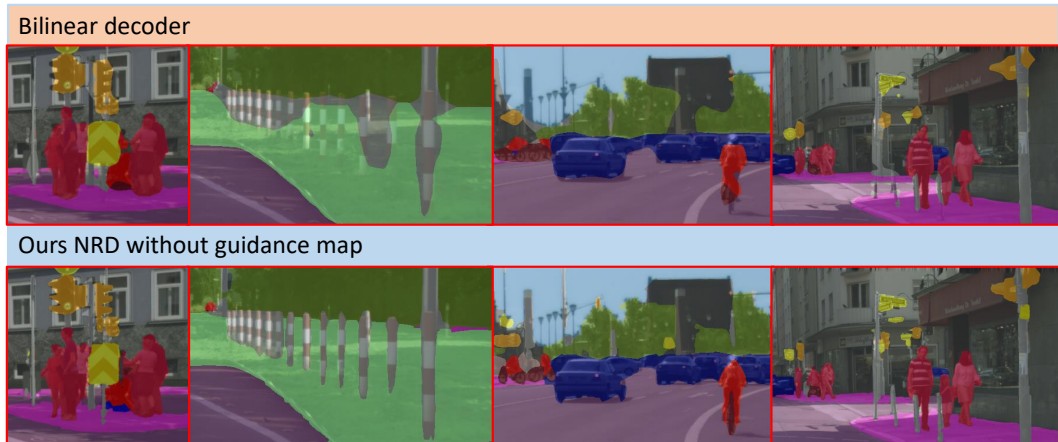

**Figure 1:** Comparison between the bilinear decoder and the NRD decoder without guidance map on the Cityscapes dataset. We can see that there is a significant improvement at the boundary region.

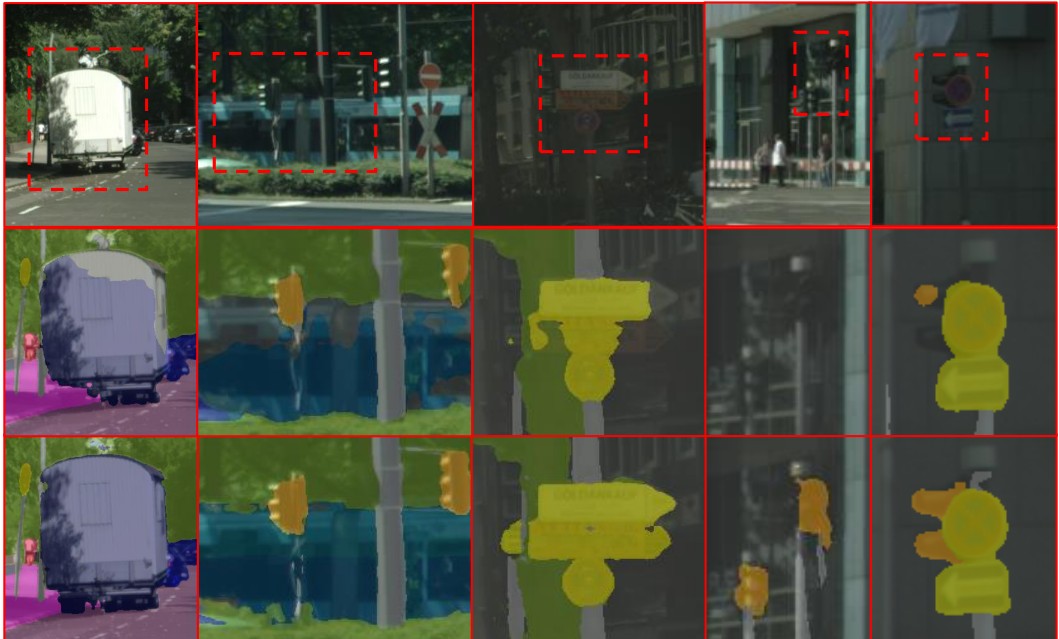

**Figure 2:** Visualization results on Cityscapes. From top to bottom: Input image; results of DeeplabV3+; and results of NRD. The single scale GFlops of these two methods are 293.6 (DeeplabV3+) vs. 234.6 (NRD). Our NRD requires less computation; yet the segmentation accuracy is superior.

of the input scale. Thus, the results can represent the effectiveness comparison between bilinear method and NRD. From the illustration we can see that it is inevitable for the decoder to lose some details during a 32 times upsampling. However, NRD clearly preserves more details than the bilinear interpolation method. These two upsampling schemes have similar computation costs.

**Comparison with the DeeplabV3+ decoder.** Fig. 2 shows more comparison between the DeeplabV3+ [3] decoder and NRD. The computational cost of the decoder part are 76.4 (DeeplabV3+) vs. 20.4 (NRD) GFlops. From the figure, we can see that in various scenes, NRD shows superior segmentation results than DeeplabV3+.

**Competitive segmentation results on ADE20K and PASCAL-Context.** Fig. 3 shows the segmentation results on ADE20K produced by NRD using ResNeXt101 backbone with multi-scale inference. We can see that in various scenes, including the bedroom, the toilet, and some outdoor scenes, NRD can generate satisfactory segmentation results. It can also perform well at boundaries such as the

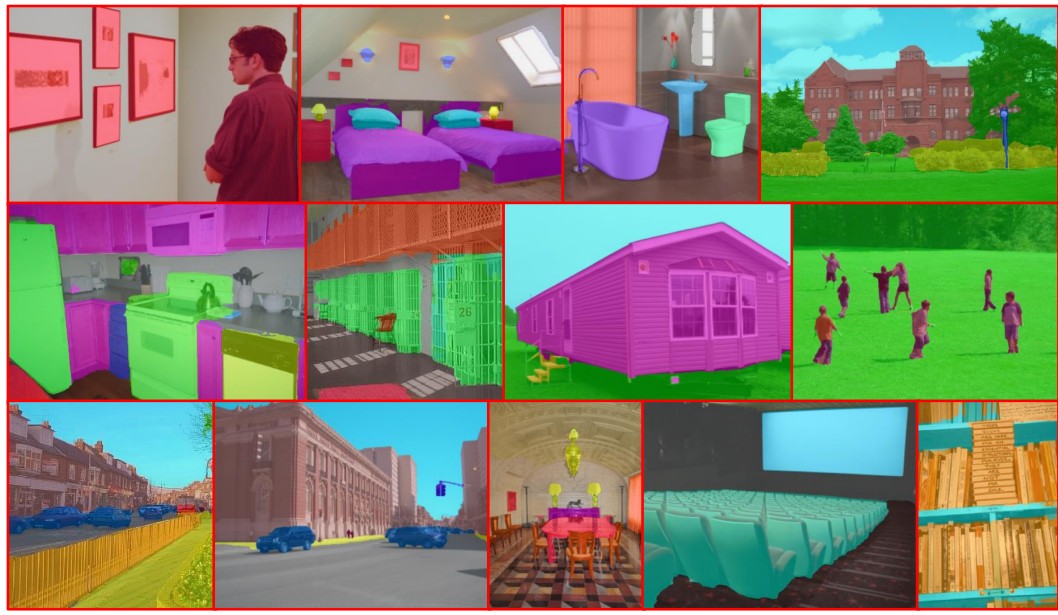

**Figure 3:** Competitive segmentation results on the ADE20K dataset. Our method performs well on various scenes, and can capture the detailed boundary's information.

human legs and the poles. Fig. 4 is the segmentation results on PASCAL-Context produced by NRD using Resnet101 backbone with multi-scale inference (60 classes).

**Detailed illustration of the NRD module.** Fig. 5 shows how an input image is processed in NRD. For the NRD structure, the guidance maps that are generated from the low level features and the coordinate maps are concatenated together. These feature maps are served as the input to the NRD structure. We attribute each patch of the feature maps with a representational network $g_{\boldsymbol{\theta}}(\cdot)$ whose parameters are dynamically generated by the controller. In Fig. 5 each block surrounded by white lines represents a patch that is processed by a particular $g_{\boldsymbol{\theta}}(\cdot)$. The result is then directly used as output of the decoder without the use of more convolutions to 'refine' the results.

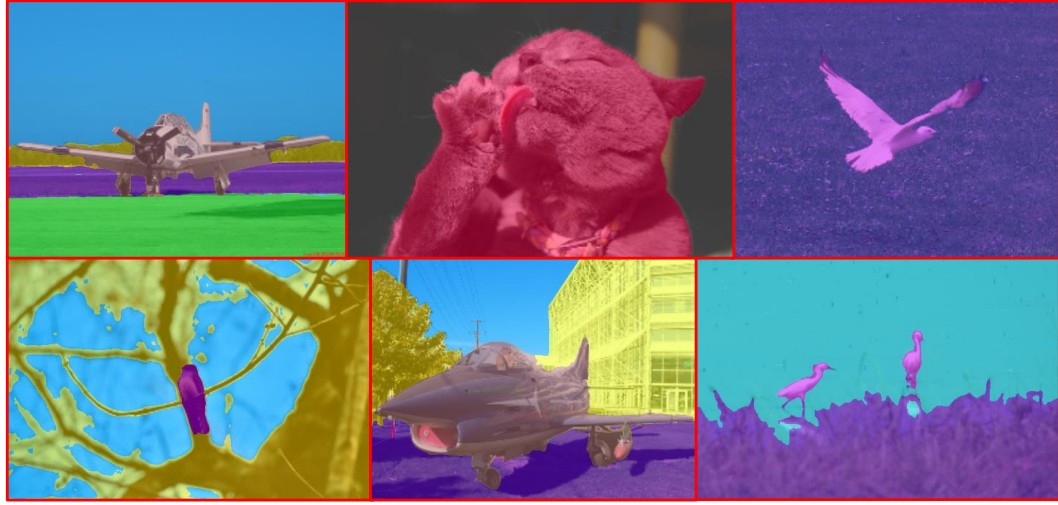

**Figure 4:** Competitive segmentation results on the PASCAL-Context dataset. The proposed method performs well on various shapes of objects.

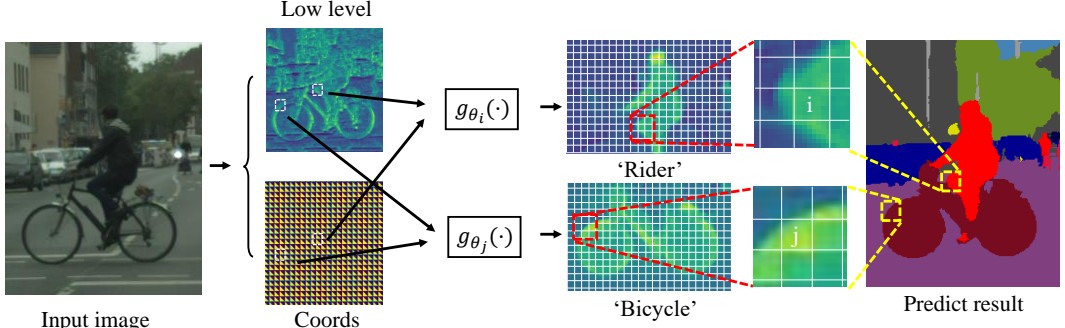

**Figure 5:** Detailed Illustration of the NRD module. Guidance maps from low-level feature maps and coordinate maps are concatenated together and pass through the representational networks $g_\theta(\cdot)$.