# OpenReview forum: "Dynamic Neural Representational Decoders for High-Resolution Semantic Segmentation"
_NeurIPS.cc/2021/Conference — NeurIPS 2021 Poster_

### Official Review · Reviewer_bjCJ · 2021-07-14

**Rating:** 6
**Confidence:** 5

**Summary:**

In this paper, the authors propose a novel decoder that is effective and compact for semantic segmentation, to recover the spatial resolutions. For the first time, we represent the local label patches using neural networks and make use of dynamic convolutions to parametrize these neural networks. The experiments show on a number of semantic segmentation benchmarks that the method is highly efficient.

**Ethical Concerns:**

no ethical concerns

**Limitations And Societal Impact:**

the limitation is presented in Main Review.

**Main Review:**

This paper propose a dynamic decoder to predict the mask for a given image. Unlike previous decoder, the proposed represents each local label patch by a compact neural network g_{\theta}with a few convolution layers interleaved with non-linearities. I think the idea is good and novel. But the paper writing and description are not clear and detailed. Many details are missing or ambiguous.

My concerns and suggestions about this paper are as follows:
   1. In line 155-160, the P is one patch from groundtruth Y'. but in line 162, a compact network $g_{\theta}$ is designed to represent the local mask patch P. So the P is the input of the network $g_{\theta}$? But the input in Fig3, the input is the guidance map and coordinate map, which is ambiguous.
   2. In line 166, at first it says the network $g_{\theta}$ is applied to coordinate map Q, but in line 170, it says the guidance map m is taken as additional input. I think it's weird to describe it like this.
   3. In line 190-191, it says "The number of output channels of the convolution is equal to the number of parameters in  $\theta$", the number of channels is C, right?    "generated parameters are then split and 191 reshaped into the weights and biases in g", what's the generates parameters' shape, how to split and reshape to weights and bias, what's the shape of weights and bias? it need to be described in detail. otherwise, it's hard to understand.
  4. in line 179, "we do not adopt this because using bilinear is sufficient when the upsampling factor is small (e.g., being 4 here).", my understanding is that the network $g_{\theta}$ is applied to coordinate map and guidance map, the coordinate map is with shape sxsx2, so how about the guidance map? please give a detailed description.
 In addition, there is no experiments for the upsampling applied or not, please add it.
   5. In table 5, 6,7, it shows DRN can achieve comparable performance with less time and computation. but in table 7, there is no more detailed comparison like DANet, HRNet, deeplabv3+. please add.






**Time Spent Reviewing:**

6

---

> ### Author Response · Authors · 2021-08-10
> **Response to Reviewer bjCJ**
>
> We thank the reviewer for his overall positive feedback and appreciation of our work. We reply individually to each raised point below.
>
>
> **1) In lines 155-160, the P is one patch from ground truth Y’. but in line 162, a compact network is designed to represent the local mask patch P. So the P is the input of the network? But the input in Fig3, input is the guidance map and coordinate map, which is ambiguous.**
>
> *A: Thanks for the suggestion. P is the target, which is one patch from the ground-truth Y’. P is the ground-truth training targets of the compact network gθ(·).  The input is the guidance maps and coordinate maps. We will improve these descriptions.*
>
> **2) In line 166, at first it says the network is applied to coordinate map Q, but in line 170, it says the guidance map m is taken as additional input.**
>
> *A: Sorry for the confusion. The motivation of this work is to recover a segmentation mask in a local patch from the coordinate map. The low-level feature map is an optional choice, which could provide low-level information to generate more details for the segmentation mask.*
>
> **3) What’s the generates parameters’ shape, how to split and reshape to weights and bias, what’s the shape of weights and bias? it needs to be described in detail. otherwise, it’s hard to understand.**
>
> *A:  Thanks for the suggestion! We will add more details and release the code. The generate parameter’s shape is 1×1×899. Please refer to line 182. The former 848 parameters are used to generate the weight for three convolutions in orders (18∗16 + 16∗16 + 16∗19). The later 51 generates the bias for three convolutions (16 + 16 + 19).*
>
> **4) In line 179, "we do not adopt this because using bilinear is sufficient when the upsampling factor is small (e.g., being 4 here).", my understanding is that the network is applied to coordinate map and guidance map, the coordinate map is with shape s×s×2, so how about the guidance map? please give a detailed description. In addition, there are no experiments for the upsampling applied or not, please add it.**
>
> *A: The coordinate map for each patch is s×s×2, and we have 1/32h×1/32w patches in total. Suppose the input image is of size(3,512,512)and s=8, then the size of the coordinate map is 128×128×2 the shape of the guidance map is 128×128×16. Therefore, the bilinear-upsampling factor could be 4. According to our experiments, use larger s to directly up-sampling 32 times will lead to larger computational costs.  Moreover, there is no significant benefit when using a larger up-sampling factor to replace the bilinear. We will add this experiment in the final version.*
>
> **5) In Tables 5, 6,7, it shows DRN can achieve comparable performance with less time and computation.  but in table 7, there is no more detailed comparison like DANet, HRNet, deeplabv3+.please add.**
>
> *A:  Thanks for the suggestion! We will add the comparison for those methods.*
>
> *NRD Experiment results on the Cityscapes test split.  ‘ms’ means that mIoU is calculated using multi-scale inference. The GFlops is measured at single-scale inference with a crop size of1024×2048. The model is trained on train Val split.*
>
> | method     | GFlops | mIoU (ms) |
> |------------|--------|-----------|
> | Deeplabv3+ | 2032.3 | 81.3      |
> | DANet      | 2214.7 | 81.5      |
> | HRNet      | 748.9  | 81.6      |
> | NRD (ours) | 390.0  | 80.5      |

---

> ### Author Response · Authors · 2021-08-28
> **Have we addressed your concerns?**
>
> Hi, do our responses answer your questions? Please let us know if you have any more questions. Thank you for your time.

---

### Official Review · Reviewer_aWGD · 2021-07-15

**Rating:** 6
**Confidence:** 3

**Summary:**

In this paper, the authors design a simple and effective decoder named NRD for the Semantic Segmentation task. They represent the local patch of the label map with a neural network, which makes it possible to restore details in the prediction with a low computational cost. Experiments on three benchmarks show the effectiveness of the proposed approach.

**Limitations And Societal Impact:**

The conclusion part in the introduction section is too redundant. The authors may consider simplifying it.

**Main Review:**

In this paper, the authors propose a computationally friendly decoder NRD. From the tables in the paper, it seems that NRD could achieve a somehow good result with a lower cost. However, the positioning of this paper is strange. On the one hand, in neither of the 3 datasets (ADE, Context, and City) does NRD achieve state-of-the-art performance. If the performance is SOTA and the computational cost is lower, it will convince me. On the other hand, the proposed method seems to be far from real-time ones and no speed metric is reported in the paper.

Besides, I have some questions about the paper.
1.	Missing comparisons. When compared with state-of-the-art, some methods are ignored. For example, SFNet [32] is listed in Table 5 and 7, but it does not show up in Table 6. According to their paper, SFNet achieves 53.8 (50.7) with ResNet-101 (ResNet-50) on Pascal Context (60 classes), which is much higher than the proposed method. For the 59-class setting, some methods even achieve more than 56 mIoU, such as OCRNet.
2.	Ablation study. In Section 3.1, when compared with DeepLabv3+, the authors do not use dilation convolutions in the backbone, resulting in output stride = 32. How about using output stride = 16 or 8? I think the ASPP module in DeepLabv3+ needs a relatively larger feature map. If the input image is 512x512, then the output feature map is 16x16 for output stride = 32.
3.	How the proposed NRD works. In table 4, only when the guidance map and the coordinate map both be used do you obtain the best performance, e.g., 79.8 mIoU. Could the authors explain how these two items improve the performance in a more detailed manner? For example, how the x-y coordinate map helps the guidance map?
4.	Implementation details. In Section 2.2, could explain how the 1/4 resolution feature map be converted to the guidance map m that has a shape of (s*s*C_m)?
5.	Line 289, the saying “better speed-accuracy tradeoff” is wrong because GFlops does not reflect the speed. You should test the speed and report the FPS number.


**Time Spent Reviewing:**

7 hours

---

> ### Author Response · Authors · 2021-08-10
> **Response to Reviewer aWGD**
>
>
> We thank the reviewer for the time and effort. We address the concerns proposed by  Reviewer aWGD in detail.
>
> **1) Missing comparisons. When compared with state-of-the-art, some methods are ignored. For example, SFNet [32] is listed in Table 5 and 7, but it does not show up in Table 6. According to their paper, SFNet achieves 53.8 (50.7) with ResNet-101 (ResNet-50) on Pascal Context (60 classes), which is much higher than the proposed method. For the 59-class setting, some methods even achieve more than 56 mIoU, such as OCRNet.**
>
> *A: The way to calculate the mIoU on Pascal Context is not consistent in the literature.  SFNet [LYZ+20] follows [ZDS+18].  They train on 59 classes to get the mIoU, and then scale the result with the factor 59/60 (confirmed with the original authors of SFNet via private communication. To our understanding, this is problematic. Thus we didn’t use this approach to evaluate). This result is reported as “evaluated on 60 classes”. If testing with the same setting as in SFNet, we achieve 55.0% mIoU, which is higher than their 53.8%.*
>
> *We employ the same setting as in HRNet [SZJ+19] and OCRNet [YCW19]. In this paper, we do not aim to achieve the best accuracy by developing a complicated model with a huge computational cost.*
> *Our main contribution is that we propose a novel approach to build a lightweight decoder for high-resolution semantic segmentation. Higher performance can be obtained if a larger backbone is employed. Ours is much more efficient than OCRNet. We achieve comparable results (54.1% vs.56%) with only one-third of the computational cost (42.9 vs. 143.9 GFlops). We also achieve better performance than OCRNet on the ADE20K Val dataset.*
>
>
> *NRD comparison for PASCAL-Context dataset using different evaluation settings*
>
> | method     | GFlops | mIoU (SFNet) | mIoU (OCRNet) |
> |------------|--------|--------------|---------------|
> | SFNet      | 90.3   | 53.8         | 52.9            |
> | OCRNet     | 143.9  | 57.2            | 56.2          |
> | NRD (ours) | 42.9   | 55.0         | 54.1          |
>
>
> **2) “When compared with DeepLabv3+, the authors do not use dilation convolutions in the back-bone, resulting in output stride = 32. How about using output stride = 16 or 8? I think the ASPP module in DeepLabv3+ needs a relatively larger feature map.”**
>
> *A: We did experiments regarding applying dilated convolutions to backbones to get stride = 16. We got slightly worse accuracy but introduces significantly heavier computational cost which disagrees with our aim to design a lightweight decoder on regular backbones.*
>
> **3) Could the authors explain how these two items improve the performance in a more detailed manner? For example, how the x-y coordinate map helps the guidance map?**
>
> *A: The dynamic parameters can be seen as a representation function describing the mask of the local patch. This representation function takes as inputs the spatial locations (and optional low-level features) and returns the desired mask results. Please refer to Figure 1 for details. Also, please refer to Table 4 for the ablation experiments. The guidance maps provide the appearance features (e.g., colors, edges, and etc.), and the coordinates maps provide the positional information.  Thus, the combination of both can result in the best result. Please also refer to SOLO v2 NeurIPS2020 which also uses dynamic Conv. with coordinate information.*
>
> **4) Could explain how the 1/4 resolution feature map be converted to the guidance map m that has a shape of (ssCm).**
>
> *A: The reviewer may have some misunderstanding of the proposed method. We do not convert the 1/4 resolution feature map to the shape of (ssCm). As we use dynamic decoders for each s×s patch(here, s=8), the feature map is divided into 1/32h×1/32w patches in total. The detailed process is as follows:*
>
> *Suppose the input image has a size of 512×512×3, then the output of the second stage of the backbone shall be 128×128×256. This output is passed through two convolution layers and the shape of the result is 128×128×Cm. That is the guidance map as mentioned. The final stage2 the output of a common backbone is 16×16×2048. We repeat an 8×8 coordinate map 16×16 times to form the final coordinate map. Then the coordinate map is of size 128×128×2 which is of the same spatial size as the guidance map. The guidance map and the coordinate map are then concatenated together for the gθ(·) to process. gθ(·) only processes its corresponding 8×8×(Cm+ 2)(Cm for guidance map and 2 for coordinate map) part of feature maps. Thus the process after the guidance map can be regarded as a group of 16×16 feature maps with their individual gθ(·) processed in parallel.*
>
> **5) better speed-accuracy trade-off, doesn’t show speed (FPS) number**
>
> *A: Thanks for the suggestion!  We test the speed of our proposed method and other competitive methods.  GFLOPs and the GPU inference time lead to similar conclusions, as shown below. ‘frames per second‘ (FPS) is reported for the speed analysis. All the methods are evaluated on an RTX3090 GPU with the single scale following the setting in MMsegmentation [MMS20]*
>
> *NRD Speed analysis for ADE20k dataset*
>
> | method       | Gflops | FPS  |
> |--------------|--------|------|
> | OCRNet       | 164.8  | 16.4 |
> | CPNet        | 286.3  | 18.3 |
> | DeeplabV3+   | 255.1  | 19.7 |
> | PSPNet       | 256.4  | 20.3 |
> | EncNet       | 218.8  | 21.9 |
> | SFNet        | 102.7  | 22.0 |
> | EFFicientFCN | 60.5   | 53.9 |
> | NRD  (ours)        | 87.9   | 34.9 |
> | NRD   (ours)       | 49.0   | 54.1 |
>
>
>
> ### reference
>
> [LYZ+20]. Xiangtai Li, Ansheng You, Zhen Zhu, Houlong Zhao, Maoke Yang, Kuiyuan Yang, Shaohua Tan, and Yunhai Tong. Semantic flow for fast and accurate scene parsing. InProc. Eur. Conf. Comp.Vis., pages 775–793, 2020.
>
> [ZDS+18]. Hang Zhang, Kristin Dana, Jianping Shi, Zhongyue Zhang, Xiaogang Wang, Ambrish Tyagi, and Amit Agrawal. Context encoding for semantic segmentation. InProceedings of the IEEE conference on Computer Vision and Pattern Recognition, pages 7151–7160, 2018.
>
> [SZJ+19]. Ke Sun, Yang Zhao, Borui Jiang, Tianheng Cheng, Bin Xiao, Dong Liu, Yadong Mu, Xinggang Wang,Wenyu Liu, and Jingdong Wang.  High-resolution representations for labeling pixels and regions.arXiv: Comp. Res. Repository, 2019.
>
> [YCW19]. Yuhui Yuan,  Xilin Chen,  and Jingdong Wang.   Object-contextual representations for semantic segmentation. InarXiv: Comp. Res. Repository, volume abs/1909.11065, 2019.
>
> [MMS20]. MMSegmentation. MMSegmentation: OpenMMLab semantic segmentation toolbox and benchmark.https: // github. com/ open-mmlab/ mmsegmentation, 2020.

---

> ### Author Response · Authors · 2021-08-28
> **Have we addressed your concerns?**
>
> Hi, have we addressed your concerns? If you have any more questions, please let us know. Thank you very much!

---

> > ### Comment · Reviewer_aWGD · 2021-09-07
> > **reply**
> >
> > My concerns have been solved. I choose to change the score from 5 to 6.

---

### Official Review · Reviewer_p8Jt · 2021-07-19

**Rating:** 6
**Confidence:** 4

**Summary:**

This paper introduces one light-weight feature decoder (NRD) for semantic segmentation. Compared with the dilated encoder, the
proposed NRD is effective for semantic segmentation due to it can be directly applied on the
standard feature encoding network. In the experimental validation, this work shows satisfied performance with much less computational cost.

**Limitations And Societal Impact:**

See above. The societal impact is shown one the last page of the manuscript.

**Main Review:**

Strengths:
The core idea of NRD firstly exploits one dynamic filter networks to generate the data-driven parameters of  the dynamic neural representational decoder. Then, the generated dynamic neural representational decoder both considers the low-level features and the coordinates to perform the feature upsampling. Then, the proposed NRD takes the advantages of the coordinates, the low-level features and the high-level features to make the dense prediction for semantic segmentation.

Weaknesses:
1). Lack of speed analysis, the experiments have compared GFLOPs of different segmentation networks. However, there is no comparisons of inference speed between the proposed network and prior work. The improvement on inference speed will be more interesting than reducing FLOPs.
2). For the detail of the proposed NRD, it is reasonable that the guidance maps are generated from the low-level feature maps. And the guidance maps can be predicted from the the first-stage feature maps or the second-stage feature maps. It is better to provide one ablation study about the effect for them.
3). Important references are missing. The GFF[1] and EfficientFCN[2] both aims to implement the fast semantic segmentation method in the encode-decoder architecture. I encourage the authors to have a comprehensive comparison with these work.

[1]. Gated Fully Fusion for Semantic Segmentation, AAAI'20.
[2]. EfficientFCN: Holistically-guided Decoding for Semantic Segmentation, ECCV'20.

**Time Spent Reviewing:**

3.5 hours

---

> ### Author Response · Authors · 2021-08-10
> **Response to Reviewer p8Jt**
>
> We thank the reviewer for his overall positive feedback and appreciation of our work. We reply individually to each raised point below.
>
> **1) Lack of speed analysis:**
>
> *A: Thanks for the suggestion!  We test the speed of our proposed method and other competitive methods.  GFLOPs and the GPU inference time lead to similar conclusions, as shown below. 'frames per second' (FPS) is reported for the speed analysis. All the methods are evaluated on an RTX3090 GPU with a single scale following the setting in MMsegmentation [3].*
>
> *NRD Speed analysis for ADE20k dataset*
>
> | method       | Gflops | FPS  |
> |--------------|--------|------|
> | OCRNet       | 164.8  | 16.4 |
> | CPNet        | 286.3  | 18.3 |
> | DeeplabV3+   | 255.1  | 19.7 |
> | PSPNet       | 256.4  | 20.3 |
> | EncNet       | 218.8  | 21.9 |
> | SFNet        | 102.7  | 22.0 |
> | EFFicientFCN | 60.5   | 53.9 |
> | NRD  (ours)        | 87.9   | 34.9 |
> | NRD   (ours)       | 49.0   | 54.1 |
>
> **2) It is better to provide one ablation study about the effect of employing different stage features as guidance maps.**
>
> *A: Thanks for the suggestion! We conduct an ablation study to get the guidance map from different stages. Results are reported on the Cityscapes Val dataset. The proposed method achieves 78.5%, 79.8%, and 78.6% with the feature from the first stage, second stage, and third stage, respectively. We employ the feature from the second stage in other experiments of this paper.*
>
> | Stage             | mIoU  |
> |-------------------|-------|
> | Stage 1           | 78.52 |
> | Stage 2 (default) | 79.8  |
> | Stage 3           | 78.64 |
>
> **3)  encourage the authors to compare with two more references about GFF [1] andEfficientFCN [2]**
>
>
> *A: Thanks for the suggestion. We will include these in the final paper.*
>
> *Results on the ADE20k dataset*
>
> | method       | backbone           | GFlops | mIoU `ms' |
> |--------------|--------------------|--------|-----------|
> | GFFNet       | dilated-Resnet-101 | >180   | 45.33     |
> | EfficientFCN | Resnet-101         | 60.5   | 45.28     |
> | NRD (ours)   | Resnet-101         | 49.0   | 45.62     |
> | NRD (ours)   | ResNeXt-101        | 87.9   | 46.09     |
>
>
>
>
> [1]. Gated Fully Fusion for Semantic Segmentation, AAAI'20.
>
> [2]. EfficientFCN: Holistically-guided Decoding for Semantic Segmentation, ECCV'20
>
> [3]. MMSegmentation. MMSegmentation: OpenMMLab semantic segmentation toolbox and benchmark.https: // github. com/ open-mmlab/ mmsegmentation, 2020.

---

> > ### Comment · Reviewer_p8Jt · 2021-08-25
> > **Thank you for your kind response**
> >
> > The responses have addressed the issues listed in the previous comment, so I will keep my rating.

---

### Decision · Program_Chairs · 2021-09-27

**Decision:**

Accept (Poster)

**Comment:**

This paper introduces a new type of feature decoder for semantic segmentation, which is more efficient than previous work. Experimental work is ok, but could have been more convincing if architectures were achieving state-of-the-art performance.

There were a number of improvements suggested by the reviewers, in particular i) improving the clarity of the paper ii) speed comparison with previous work which should be included in the final version of the paper.